# Genomics of MPNST (GeM) Consortium: Rationale and Study Design for Multi-Omic Characterization of NF1-Associated and Sporadic MPNSTs

**DOI:** 10.3390/genes11040387

**Published:** 2020-04-02

**Authors:** David T. Miller, Isidro Cortés-Ciriano, Nischalan Pillay, Angela C. Hirbe, Matija Snuderl, Marilyn M. Bui, Katherine Piculell, Alyaa Al-Ibraheemi, Brendan C. Dickson, Jesse Hart, Kevin Jones, Justin T. Jordan, Raymond H. Kim, Daniel Lindsay, Yoshihiro Nishida, Nicole J. Ullrich, Xia Wang, Peter J. Park, Adrienne M. Flanagan

**Affiliations:** 1Division of Genetics and Genomics, Boston Children’s Hospital, Boston, MA 02115, USA; katherine.piculell@childrens.harvard.edu; 2European Molecular Biology Laboratory, European Bioinformatics Institute, Hinxton, Cambridge CB10 1SD, UK; icortes@ebi.ac.uk; 3Department of Pathology, University College London Cancer Institute, Bloomsbury, London WC1E 6BT, UK; n.pillay@ucl.ac.uk (N.P.); a.flanagan@ucl.ac.uk (A.M.F.); 4Royal National Orthopaedic Hospital, Brockley Hill, Stanmore, Middlesex HA7 4LP, UK; daniel.lindsay1@nhs.net; 5Oncology Division, Department of Medicine, Washington University School of Medicine in St. Louis, St. Louis, MO 63110, USA; hirbea@wustl.edu; 6Department of Pathology, New York University Langone Health, New York City, NY 10016, USA; matija.snuderl@nyulangone.org; 7Department of Pathology, Moffitt Cancer Center, Tampa, FL 33612, USA; marilyn.bui@moffitt.org; 8Department of Pathology, Boston Children’s Hospital, Boston, MA 02115, USA; Alyaa.Al-Ibraheemi@childrens.harvard.edu; 9Department of Pathology and Laboratory Medicine, Mt. Sinai Hospital, Toronto, ON M5G 1XF, Canada; Brendan.Dickson@sinaihealth.ca; 10Department of Pathology, Lifespan Laboratories, Rhode Island Hospital, Providence, RI 02903, USA; jhart5@lifespan.org; 11Departments of Orthopaedics and Oncological Sciences; Huntsman Cancer Institute, University of Utah, Salt Lake City, UT 84112, USA; kevin.jones@hci.utah.edu; 12Pappas Center for Neuro-Oncology, Massachusetts General Hospital, Boston, MA 02114, USA; jtjordan@mgh.harvard.edu; 13Department of Medical Oncology, Princess Margaret Cancer Center, University Health Network, Toronto, ON M5G 2C1, Canada; raymond.kim@uhn.ca; 14Department of Rehabilitation, Nagoya University Hospital, Nagoya 466-8550, Aichi, Japan; ynishida@med.nagoya-u.ac.jp; 15Department of Neurology, Boston Children’s Hospital, Boston, MA 02115, USA; nicole.ullrich@childrens.harvard.edu; 16GeneHome, Moffitt Cancer Center, Tampa, FL 33612, USA; xia.wang@moffitt.org; 17Department of Biomedical Informatics, Harvard Medical School, Boston, MA 02115, USA; peter_park@hms.harvard.edu

**Keywords:** genomics, MPNST, tumor evolution, neurofibromatosis, pathology, next generation sequencing, clinical genetics

## Abstract

The Genomics of Malignant Peripheral Nerve Sheath Tumor (GeM) Consortium is an international collaboration focusing on multi-omic analysis of malignant peripheral nerve sheath tumors (MPNSTs), the most aggressive tumor associated with neurofibromatosis type 1 (NF1). Here we present a summary of current knowledge gaps, a description of our consortium and the cohort we have assembled, and an overview of our plans for multi-omic analysis of these tumors. We propose that our analysis will lead to a better understanding of the order and timing of genetic events related to MPNST initiation and progression. Our ten institutions have assembled 96 fresh frozen NF1-related (63%) and sporadic MPNST specimens from 86 subjects with corresponding clinical and pathological data. Clinical data have been collected as part of the International MPNST Registry. We will characterize these tumors with bulk whole genome sequencing, RNAseq, and DNA methylation profiling. In addition, we will perform multiregional analysis and temporal sampling, with the same methodologies, on a subset of nine subjects with NF1-related MPNSTs to assess tumor heterogeneity and cancer evolution. Subsequent multi-omic analyses of additional archival specimens will include deep exome sequencing (500×) and high density copy number arrays for both validation of results based on fresh frozen tumors, and to assess further tumor heterogeneity and evolution. Digital pathology images are being collected in a cloud-based platform for consensus review. The result of these efforts will be the largest MPNST multi-omic dataset with correlated clinical and pathological information ever assembled.

## 1. The Complex Genomic Landscape of MPNST (Malignant Peripheral Nerve Sheath Tumors)

Malignant Peripheral Nerve Sheath Tumors (MPNST) confer high morbidity and currently has limited treatment options. Patients with neurofibromatosis type 1 (NF1) have an 8–13% lifetime risk of developing malignant peripheral nerve sheath tumor (MPNST) which is the most frequent cause of early death [1]. Surgical resection with negative margins is the principal curative therapeutic modality, but is not always feasible [2,3]. Radiation and/or chemotherapy are often used in the adjuvant or neoadjuvant setting. As there are no randomized trials for MPNST to justify this treatment, recommendations are based on data from the high grade soft tissue sarcoma group as a whole, including both sporadic and NF1-associated MPNST [4,5,6]. The 5 year overall survival rate is modest, with one meta-analysis estimating survival at only 26–39%, and with high rates of metastasis, morbidity, and mortality [2]. In the setting of metastatic disease, treatments are limited to palliative chemotherapy and clinical trials [7,8]. Despite a recent increase in the knowledge of molecular aberrations underpinning MPNST there have not been any new effective therapeutic options developed; this may be explained by the rarity of these tumors. 

Prior studies assessing the MPNST genomic landscape have been relatively small. This consortium therefore sees an opportunity to expand upon this foundation of knowledge. Somatic loss of either *TP53* or *CDKN2A* has been demonstrated in essentially all MPNST, either via somatic copy number alterations (SCNAs) or single nucleotide variants (SNVs) [9]. The genetic complexity of MPNST was better understood after two independent studies highlighted the prominent role of Polycomb repressor complex 2 (PRC2) inactivation in the development of MPNST through somatic inactivating mutations or deletions in *SUZ12, EED or EZH2* [10,11]. A subset of MPNSTs with PRC2 loss shows loss of trimethylation at lysine 27 of histone H3 (H3K27me3) [10]. This consortium will leverage a larger sample size to determine what proportion of MPNST show PRC2 loss, and how that may correlate with other aspects of the data. 

This consortium also represents an opportunity to expand upon prior studies demonstrating altered methylation and gene expression patterns in MPNST development. For example, a project on soft-tissue sarcomas conducted by The Cancer Genome Atlas (TCGA) performed multi-omic analysis of over 200 sarcoma specimens (*n* = 5 MPNST) found differential patterns of methylation and gene expression in certain sarcoma types [9]. Additional studies of altered gene expression in MPNST have been recently reviewed [12]. Similarly, a methylation classifier analysis of 171 peripheral nerve sheath tumors that included 28 conventional high-grade MPNST, six atypical neurofibromas and other related tumors, such as neurofibromas and schwannomas, demonstrated patterns that helped differentiate the high grade tumors [13]. More specifically, by unsupervised hierarchical clustering, atypical neurofibromas and low-grade MPNST were indistinguishable, and also harbored frequent *CDKN2A* deletions. High-grade MPNST formed two distinct methylation groups which shared a frequent loss of the *NF1* locus, and also showed some differences based on anatomical location. This highlights the potential emerging role of DNA methylation profiles for diagnosis and categorization of nerve sheath tumors. Our consortium is optimistic that further exploration of the role of epigenetics, and related alterations in gene expression, in the pathogenesis of MPNST will prove fruitful. In support of this view, a recent report showed that methylated *RASSF1A* in MPNSTs identified patients with *NF1* silencing and an inferior prognosis, suggesting that methylation at a specific locus may correlate with clinical behavior [14].

## 2. Knowledge Gaps in the Understanding of Tumor Drivers and Evolution of MPNST

Prior studies have identified a number of recurring molecular events that appear in a majority of MPNSTs, but there is lack of uniformity of any of these molecular markers across all tumors in this histological group. Our consortium thinks that this knowledge gap is due primarily to the low overall incidence of MPNST, and therefore relatively low numbers of viable samples included in prior studies. For example, NF1-associated MPNSTs typically arise by malignant transformation of an existing plexiform, or nodular or atypical/ANNUBP neurofibroma [9]. The development of plexiform neurofibromas (PN) follows Knudsen’s two-hit hypothesis with loss of heterozygosity of the *NF1* tumor suppressor gene, the likely initiating rate-limiting event for tumorigenesis [15]. However, loss of function of the second *NF1* allele has not been observed in all specimens studied, suggesting that other mechanisms are likely important in PN development [16]. Since the *NF1* gene is large, a second genetic event affecting *NF1* may not always be easy to detect [16]. 

The genomic landscape is more complex for MPNSTs, as compared to PN and ANNUBP; this is reflected in the acquisition of additional mutations, genomic rearrangements and copy number alterations (CNA) as the histological appearance of the tumor progresses [17]. Whole genome sequencing is expected to provide the best opportunity to detect these types of mechanisms, but only a small number of MPNST whole genomes from patients with NF1 have been published and no pathognomonic chromosomal translocations have been identified [10,11,18]. Collectively, these studies identified frequent somatic loss of *NF1*, *CDNKN2A*, *TP53*, and genes from the PRC2 complex, specifically *SUZ12* and *EED*. A variety of other genes have been implicated in the progression from benign to malignant peripheral nerve sheath tumors, including candidate driver genes such as *EPC1*, *CHD4*, *AEBP2* and *ATRX.* These genes have been implicated in MPNST primarily because of their critical interaction with molecules in the PRC2 complex [11]. Other research has implicated additional pathways (e.g., Hippo/LATS), but the relative contributions of these in the pathogenesis of the disease is not well understood, perhaps due to the relatively low number of samples studied overall [19]. Furthermore, copy number alterations (CNA) on several chromosomes have been identified through array comparative genomic hybridization (aCGH) studies, and these have been reviewed in detail elsewhere [12,17]. 

Intra-tumor heterogeneity is also a challenge to understanding the molecular drivers of tumorigenesis and disease progression in MPNST [17,20]. There is also substantial interpatient tumor heterogeneity. These findings highlight that such genetic variability within and between tumors plays a critical role in clinical management and treatment resistance. Hence, there is a need to catalogue the molecular events in the primary tumor and understand how these change over time and with treatment (e.g., mutations acquired during chemotherapy that lead to drug resistance). 

## 3. Establishment of the Genomics of MPNST Consortium to Address Knowledge Gaps

Our overarching goal for the GeM(Genomics of MPNST) Consortium is to accelerate the identification of diagnostic and prognostic markers, and potential therapeutic targets for MPNST, through comprehensive molecular profiling of these rare tumors and international sharing of clinical and genomic datasets across multiple institutions worldwide. A consortium-based approach was deemed necessary in order to facilitate collection of a sufficiently large number of samples. Consequently, the Genomics of MPNST (GeM) Consortium was initiated in 2017 by the NF Research Initiative (NFRI), a philanthropically-funded translational research program at Boston Children’s Hospital. The overarching focus of the GeM Consortium is to facilitate the collection and sharing of molecular and clinical data on rare NF1-related malignant and pre-malignant tumors related to MPNST, such as atypical neurofibromatous neoplasms of uncertain biological potential (ANNUBP), and, as a lower priority, sporadic MPNST, with the goal of facilitating more rapid progress in translational research to improve clinical outcomes. 

Several factors influenced our decision to pursue a genomics project related to MPNST. We were motivated by the energy of a small but dedicated interdisciplinary community of experts with enthusiasm for pursuing genomics as a mechanism to identify potential therapies. This was manifested by our effort to pursue the aim set out at a 2001 MPNST consensus conference “to establish an international, multidisciplinary consortium of experts on MPNST and NF1, to collate the known clinical and genetic information about these tumors and to establish a database to record information in a uniform manner” [7]. The desire to accomplish that goal was reignited at the 2016 MPNST “State of the Science” meeting at the National Cancer Institute, in which some current GeM Consortium members participated, and led to the formation of the current GeM Consortium [15].

The GeM Coordinating Center began recruiting collaborators internationally in the Summer of 2017 through promotion on the NFRI website. In addition, a request for applications distributed at the Children’s Tumor Foundation’s annual meeting in June and by email to members of the NF research community. By the Fall of 2017, 13 founding member institutions, representing five countries, had established the GeM Consortium. Ultimately, four sites had to withdraw due to inability to provide the required specimens. Each site nominated one representative from their institution to serve on the GeM Steering Committee (SC), a multidisciplinary group that provides oversight to all aspects of this collaborative effort. The GeM Steering Committee established three Working Groups, composed of SC members and experts from their respective sites, to address logistical issues related to the following Consortium functions: Genomics and Informatics, Oncology and Pathology, and Data Use and Publications. This organizational structure allows for equal representation from all GeM member institutions, and multidisciplinary input into the creation of policy and research strategy.

## 4. Specimen and Clinical Data Collection, and Specimen Processing

The GeM Consortium Coordinating Center at Boston Children’s Hospital and Dana-Farber/Harvard Cancer Center established a non-human subjects research protocol to allow for the aggregation and analysis of de-identified clinical and genetic data, and specimens from GeM collaborators collected under pre-existing IRB-approved protocols at each participating institution. These pre-existing IRB protocols already permitted specimen and data collection, sharing with outside investigators, and comprehensive molecular testing. We also explored the possibility of establishing a central IRB for prospective collection, but collecting specimens via existing IRB protocols was preferable because MPNST is a rare tumor, and prospectively collecting enough samples would not have been possible in the 1-2 years allotted for establishing this collection. For example, the samples aggregated for our consortium were collected over a span of almost 20 years, indicating that it would have taken approximately that long to establish a similar sample size through prospective collection. 

Specimens sent to the Coordinating Center include MPNST and related neurofibroma as fresh frozen, paraffin-embedded tissue, tissue microarrays, or isolated DNA/RNA along with paired normal samples such as peripheral nerve or blood. Collection and processing of specimens, followed by nucleic acid extraction, was coordinated among four Pathology Departments selected from the participating sites (Boston Children’s Hospital, Moffitt Cancer Center, Mt. Sinai Hospital Toronto, and University College London). For comprehensive molecular analyses, the first step included pathological analysis of sectioned, fresh frozen tumor specimens to select the most viable areas from high quality tumor samples judged by cellularity, lack of necrosis and areas with little contaminating non-neoplastic tissue. The GeM Consortium’s Standard Operating Procedure (SOP) for tissue processing, pathology review and molecular analysis is modelled on the Royal National Orthopaedic Hospital’s SOP for the 100,000 Genomes Project, founded by England’s National Health Service in 2012. Additional tissue sections from the regions selected for multi-omic analysis were collected in order to perform immuno-histochemical classification. In addition, whole slide digital pathology images have been collected to facilitate a cloud-based histology review and correlation with molecular markers in tissue sections.

The GeM Consortium partnered with the international MPNST Registry at Washington University School of Medicine (WUSM) for the collection of comprehensive clinical data and diagnostic imaging reports. All data are collected and housed in REDCap (https://www.project-redcap.org/). This worldwide database collects the clinical data in a comprehensive and standardized manner for each participant from diagnosis forward. Data include demographic information, disease course, tumor size/anatomical location, histological/immuno-histochemical characteristics, diagnostic imaging, surgical procedures, systemic treatment information, neoadjuvant therapy, toxicity, clinical outcomes and survival. Logistic regression models will be used to correlate clinical outcome with MPNST features. A summary of important clinical variables associated with the collected tumor specimens is presented in Table 1.

We ultimately collected 96 fresh frozen MPNST (60 NF1-related MPNST and 36 non-NF1 or unknown) with paired normal specimens (i.e., peripheral blood) from 86 subjects (51 with confirmed NF1 diagnosis; 35 non-NF1 or unknown) (Figure 1). The size of our final cohort was more influenced by the availability of viable biological specimens as compared to the availability of detailed clinical data. For example, when we initially established the consortium, there were 14 participating sites, estimating that approximately 215 unique MPNST specimens would be available for study. Subsequently, three sites had to withdraw due to inability to obtain permission from their home institution to share samples. Further, there was one site that had to withdraw due to nonviable samples. Among the ten active sites, we collectively estimated that there would be 165 MPNST specimens. Unfortunately, several samples were nonviable, either due to low DNA quantity or quality. Ultimately, our original estimate of 215 tumors decreased to 165 tumors after four sites withdrew, and then decreased further to 96 tumors after accounting for samples that were nonviable due to either low DNA quantity or quality.

## 5. Plan for Multi-Omic Characterization of MPNST

### 5.1. Phase 1: Multi-Omic Profiling on Frozen Tumor Material to Study MPNST Genomic Complexity, Tumor Drivers, and Tumor Heterogeneity

Based on the aforementioned expert Pathology review, these tumors had a high, but variable, purity of approximately 40–70%. Whole genome sequencing (WGS) is being performed on all fresh frozen tumor samples at 80x coverage using libraries created with the TruSeq DNA-PCR free kit (Illumina Inc., San Diego, CA, USA). Given the known tumor heterogeneity and variable purity, we elected to perform deeper sequencing than previously reported to identify drivers of tumorigenesis. WGS will also be performed on paired normal germline DNA samples at 30x coverage. Although it requires more DNA, a PCR-free library preparation was selected to minimize artifacts of both single nucleotide variants and copy number variants that may arise from PCR amplification, and this is particularly important for the tumor-derived DNA. In addition, each of these bulk frozen tumor specimens is being analyzed by RNAseq with the TruSeq Transcriptome kit (Illumina Inc., San Diego, CA, USA) and epigenetic profiling with DNA methylation analyses using the Illumina EPIC array platform (Methylation EPIC 850k BeadChip). Control data for RNAseq analysis has been collected on the same platform from nine samples of healthy peripheral nerve frozen tissue collected from a subset of individuals in this cohort. A smaller number of high quality samples derived from snap frozen DNA will undergo whole genome bisulfite sequencing to achieve a genome-wide view of DNA methylation with a higher resolution compared to the EPIC array. Finally, multi-regional sampling was also performed on a subset of nine fresh frozen NF1-related MPNST specimens to assess intra-tumor variation (i.e., tumor heterogeneity) by performing 500× exome sequencing, RNA-seq, and epigenetic profiling.

### 5.2. Phase 2: Extensive Characterization of Tumor Heterogeneity and Evolution Using FFPE MPNST Samples from Phase 1 and Additional Informative Cases

To study intra-tumor (i.e., heterogeneity) and inter-tumor (i.e., evolution) variability, the GeM Consortium will conduct additional molecular analyses on formalin-fixed paraffin embedded (FFPE) MPNST specimens from pathology archives. The first priority will be to analyze FFPE specimens from tumors and subjects represented in the first phase of the project (WGS on fresh frozen tumor and normal DNA). The consortium will also identify potentially more informative cases where there are multiple tumors from the same person (e.g., an MPNST and precursor lesion such as ANNUBP or metastases), and tumors within which there is more than one line of differentiation (for example, nerve sheath and osteosarcomatous differentiation) with the hope that multi-regional sampling from multiple FFPE tumors representing differences in space and changes over time will uncover how and why precursor neurofibromas evolve into MPNST.

Having an accurate characterization of tumor type is critically important for downstream data analysis. GeM pathologists use the World Health Organization’s (WHO’s) histopathological and morphological classification system to establish tumor diagnosis and determine grade of MPNST, and also to confirm the diagnosis of any related precursor tumors that are available from subjects with a MPNST [21]. Immunohistochemistry studies based on tissue microarray (TMA) will add further detail to the classification of NF1-related MPNST, sporadic MPNST, and related tumors such as ANNUBP and PN. Multi-regional tumor cores (*n* = 5) from each FFPE tumor will be selected, on the basis of histological features, such as rhabdomyosarcomatous, angiosarcomatous, osteosarcomatous areas, etc. DNA will be extracted from these regional samples and will be subjected to deep (500×) bulk whole exome sequencing (WES), copy number analysis using Illumina’s Omni Array and DNA methylation analysis using Illumina’s EPIC array. Two adjacent cores of tumor will be taken for building a TMA. Annotation and analysis of digital images of FFPE slides will be used to collate pathological features and clinical outcome data with genomic data.

### 5.3. Bioinformatics Analysis for Multi-Omic Profiling of MPNSTs

Although an exhaustive list of all intended analyses would be beyond the scope of this white paper, we would like to highlight a few key areas of interest. GeM Consortium data analysis pipelines will utilize the alignment and mutation detection pipelines used by recent national and international cancer genomics consortia, such as methods for WES, WGS, and RNAseq used in the Pan-Cancer Atlas Consortium and in the Pan-Cancer Analysis of Whole Genomes (PCAWG) project [22]. This will include harmonization of third-party contributed data by lifting over to common reference, re-alignment, re-calling variants, and/or re-quantitating RNA-seq data. Our GeM analysis team has extensive experienced with these methods [23]. Uniform quality-control, alignment and processing of the WGS, WES, and RNA-seq will be performed, including the detection of somatic single-nucleotide variants, small insertions and deletions, microsatellite instability, structural variants, copy number variants, and gene fusions using WGS/WES data. Neoadjuvant treatment status will be considered during the bioinformatic analysis.

Analysis pipelines will also include gene expression quantification and gene fusion detection using RNAseq data. Epigenomic alterations will be correlated with other data types. Our analysis will also include integration of data generated outside the GeM Consortium, re-processing the data as needed. Mutational signature analysis will be performed on this dataset. Mutational signatures have emerged as a useful computational approach for identification of the biological processes that generate somatic mutations. Mutational signatures refer to patterns of nucleotide changes and their contexts, occurring due to various environmental carcinogens or endogenous DNA damage processes. A good example of this is the work of Alexandrov et al. that demonstrated tobacco induces a specific pattern of C>A transversions during the lifetime of the lung cancer cell [24]. Although these patterns can indicate specific etiological processes, they can also serve as markers of immunological response as seen in breast cancer where DNA damage response signatures were associated with lymphocytic infiltration. 

There is also emerging evidence that extraction of copy number signatures from a common cancer, such as high grade serous ovarian carcinoma, provides more robust prognostic information than pathological grading or the analysis of single gene mutations [25]. In sarcomas, copy number signatures have proved useful in understanding the evolutionary trajectories of the genomically complex cancers [26]. In a meta-analysis study of >5000 samples of 12 cancer types, some patterns of somatic copy number alterations were associated with reduced expression of cytotoxic immune signatures [27]. Moreover, the copy number scores were predictive of response to immune checkpoint blockade. There is also some preliminary evidence from sequencing of osteosarcoma genomes that a homologous recombination deficiency signature may be a consistent feature of that tumor type. A similar signature of “BRCAness” in breast cancer is predictive of sensitivity to poly (ADP-ribose) polymerase (PARP) inhibitors [28]. These are just some examples of the type of analysis that will be possible with this large dataset.

### 5.4. GeM Data Display and Availability 

The goal of the GeM Consortium, like that of the Genomics Evidence Neoplasia Information Exchange (GENIE) effort by the American Association of Cancer Research (AACR), is to enable identification of novel therapeutic targets and genomic markers of response to therapy [29]. To do so, we are developing a secure Django-based web platform for the visualization and reporting of the data generated by the GeM Consortium. De-identified molecular data derived from submitted specimens and clinical data submitted by GeM sites will be hosted in this database and may be accessed by sites and other qualified researchers from the broader community under the terms set by the Data Use and Publications Working Group and with approval of the Steering Committee. We will create a publicly accessible instance of cBioPortal to display molecular alterations that resulted from our multi-omic analysis of MPNSTs [30,31].

## 6. Concluding Remarks and Future Directions

The GeM Consortium’s ultimate goal is to improve clinical care for patients with MPNST through a better understanding of genetic and epigenetic drivers of MPNST initiation and progression. We are motivated by our desire to provide the best care for our patients. Collectively, GeM clinicians and their multi-disciplinary teams care for approximately 2400 patients with NF1 per year. Due to the appreciable incidence of MPNST in this patient population, a better approach to treatment is needed, and we think that this will only be possible through a better understanding of the molecular drivers of MPNST development and progression. Although variation in genes such as *NF1*, *CDKN2A/B*, *TP53*, and *SUZ12* and/or *EED* are found in most MPNST, the precise order and timing of these changes remain poorly described, and this may be important for understanding the early stages of tumor development and/or the development of treatment resistance. 

A comprehensive mutational, rearrangement and copy number signature analysis has not been performed in MPNST. The patients recruited through the consortium and their samples will serve as a valuable resource to facilitate the identification of such signatures through mRNA expression analysis, epigenetic profiling, and the correlation with clinical endpoints. Our hope is that this dataset will provide the most comprehensive knowledge to date and reveal previously unrecognized important pathways for MPNST initiation and progression. Beyond the currently described approach, future efforts are likely to include single-cell DNA and RNA analyses to describe better clonal drivers within MPNSTs. We hope that the knowledge gained through the GeM Consortium’s efforts will inform both pre-clinical studies of MPNST and selection of candidate drugs for future clinical trials. For more information or to contact us, please visit www.NFResearch-Childrens.org. 

## Figures and Tables

**Figure 1 genes-11-00387-f001:**
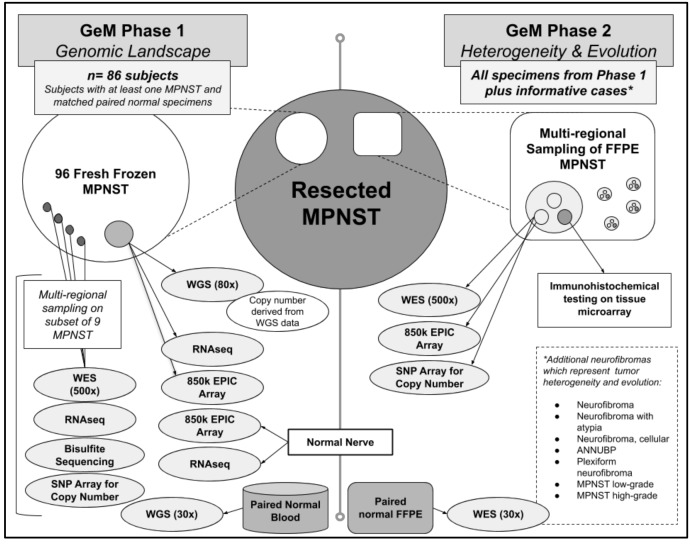
The GeM(Genomics of MPNST) Consortium will conduct multi-omic analyses on both fresh frozen and paraffin-embedded tissue samples of resected MPNSTs(Malignant Peripheral Nerve Sheath Tumors) and related neurofibroma, and normal nerve.

**Table 1 genes-11-00387-t001:** Clinical Variables of Fresh Frozen MPNST Collected by GeM Consortium.

	NF1-Related	Sporadic or Unknown Diagnosis	Total MPNST
	*n* (%)	*n* (%)	*n* (%)
Fresh Frozen MPNST	60 (62.5%)	36 (39.6%)	96 (100%)
**Tumor Grade**			
Low Grade	5 (5.2%)	2 (2.1%)	7 (7.3%)
High Grade	49 (51.0%)	32 (33.3%)	81 (84.4%)
Unknown	6 (6.3%)	2 (2.1%)	8 (8.3%)
**Neo-Adjuvant Treatment**			
Chemotherapy	8 (8.3%)	5 (5.2%)	13 (13.5%)
Radiation	3 (3.1%)	9 (9.4%)	12 (12.5%)
Chemotherapy and Radiation	2 (2.1%)	2 (2.1%)	4 (4.2%)
No neo-adjuvant treatment	44 (45.8%)	20 (20.8%)	64 (66.7%)
Unknown	3 (3.1%)	0 (0%)	3 (3.1%)
**Tumor Anatomic Location**			
Head/Neck/Face	4 (4.2%)	0 (0%)	4 (4.2%)
Lower Limb	18 (18.8%)	18 (18.8%)	36 (37.5%)
Upper Limb	14 (14.6%)	12 (12.5%)	26 (27.1%)
Brachial Plexus	3 (3.1%)	1 (1.0%)	4 (4.2%)
Lumbosacral Plexus	5 (5.2%)	3 (3.1%)	8 (8.3%)
Trunk	8 (8.3%)	1 (1.0%)	9 (9.4%)
Retroperitoneum	1 (1.0%)	0 (0%)	1 (1.0%)
Other	7 (7.3%)	1 (1.0%)	8 (8.3%)
Total MPNST	60 (62.5%)	36 (37.5%)	96 (100%)

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
