# Peer review of "Genomics of MPNST (GeM) Consortium: Rationale and Study Design for Multi-Omic Characterization of NF1-Associated and Sporadic MPNSTs"

_genes, 2020, doi:10.3390/genes11040387_

Round 1

Reviewer 1 Report

The authors have satisfactorily answered my concerns - this is an excellent and necessary project.

Reviewer 2 Report

The article is clear and well written, but there are some points that could be enhanced:

-             Even if there are few conditions in which genotype-phenotype correlations have been found, these cases should be mentioned. The evidences, gathered to date, could be denied or confirmed by the new genomic technologies. The Authors should include this aspect.

-             A better understanding of genetic and epigenetic drivers of the MPNST initiation and progression, could have a possible impact on prenatal screening and identification of prognostic risk. The Authors shoud develop this issue;

-             The results of this study may contribute to give further insights into the clinical and genomic characterization of the disease. A successful approach is based on a comprehensive (multiprofessional/polyspecialistic) and individualized management of the patient, from birth (and even before) to adulthood. This important topic should be developed by the Authors.

This manuscript is a resubmission of an earlier submission. The following is a list of the peer review reports and author responses from that submission.

Round 1

Reviewer 1 Report

Thanks to Dr. Miller and colleagues fo this highly ambitious project. This manuscript essentially serves as a white paper declaring the purpose and methods of this project. Extremely powerful methods are being employed, so the likelihood of discovery of unique findings is higher, although overall numbers are low enough that true significance may not be achieved. Likewise, the source of the tumors to be investigated is not well elucidated, so those samples from tumors that have been previously treated, especially those tumors that have been previously radiated, may generate excess "noise" that may be problematic, especially given the deep analytic dive that is intended. 

Certainly this project has the potential to be the primary reference for understanding MPNST for years to come. For those like me who are making efforts to elucidate the biology and potential treatment targets for this dangerous malignancy, I am hopeful for data generation that will be helpful to this patient population.

Reviewer 2 Report

This is a manuscript describing the initial development of a consortium that has collected MPNST specimens, and aims to completely profile them.  It is actually schizophrenic: on the one hand, it provides an excellent. comprehensive, and well written review of the MPNST literature, which is valuable.  The description of the organizational structure is also relevant. 

On the other hand, the abstract does not focus on the review aspect of the paper.  There is no data in the paper.  If the abstract is completely rewritten to emphasize that it is a review, with minor consideration given to the consortium development and its successes/failures, perhaps the manuscript could be most relevant.

Reviewer 3 Report

The description of the consortium and intended studies are outstanding!  Unfortunately, there are no results.  Hence, the title is misleading as multi-omics characterization is the plan, but has not occurred.  I eagerly anticipate the results.  Lines 183-4 mention preliminary data, but none is presented.  

While the Intro serves as a bit of a review, the whole genomes of MPNSTs that are mentioned in lines 151-152 are not referenced or described.  Further, there have been at least a handful of papers published regarding differential gene expression in MPNSTs, but these have been largely ignored despite the fact that RNASeq is one of the goals for all tumor samples.  

Mention is made of utilizing blood and adjacent tissues (line 209) as controls/comparator samples, but what is the appropriate control for RNASeq and epigenetic profiling?  Tumor and blood RNA profiles aren't comparable.  Please elaborate.

Reviewer 4 Report

This well written manuscript describes a comprehensive multi-omic sequencing approach to NF1-related MPNSTs. It goes into great depth describing the makeup and function of the research consortium, as well as the strong rationale for tissue collection. This project will be extremely impactful once the data is collected, reported and made publicly available. Unfortunately, there was no data presentation(?), so it's difficult to be enthusiastic about the manuscript as it is currently written. If there is more data available to the reviewers, perhaps it should be resubmitted once the project is more mature and scientific conclusions can be made. I want to encourage the authors to resubmit because of the tremendous impact this paper will have on the NF field once it's complete.